# Evaluation of Body Composition and Biochemical Parameters in Adult Phenylketonuria

**DOI:** 10.3390/nu16193355

**Published:** 2024-10-02

**Authors:** Mehmet Cihan Balci, Meryem Karaca, Dilek Gunes, Huseyin Kutay Korbeyli, Arzu Selamioglu, Gulden Gokcay

**Affiliations:** Division of Nutrition and Metabolism, Istanbul Faculty of Medicine, Children’s Hospital, Istanbul University, 34093 Istanbul, Turkey; meryem.karaca@istanbul.edu.tr (M.K.); drdilekgunes@gmail.com (D.G.); kutaykorbeyli@hotmail.com (H.K.K.); arzuceyln@hotmail.com (A.S.); ghuner@istanbul.edu.tr (G.G.)

**Keywords:** phenylketonuria, overweight, obesity, body composition, body mass index

## Abstract

Background/Objectives: Phenylketonuria is a hereditary metabolic disorder characterized by a deficiency of phenylalanine hydroxylase. The main treatment for PKU is a phenylalanine-restricted diet. The exclusion of protein rich natural foods and inclusion of low-Phe substitutes may give rise to an imbalanced diet, and the increased risk of overweight and obesity in PKU is a cause for concern. We aimed to evaluate the body composition and nutritional biochemical biomarkers in adult PKU patients who are on Phe-restricted and essential amino acid-supplemented nutrition therapy and to investigate the relationships between these parameters and patient gender, adherence to dietary therapy, and disease type, defined as mild or classic PKU. Methods: The study group comprised 37 PKU patients and 26 healthy siblings as controls. The participants were assessed based on an analysis of anthropometric parameters, body composition, and biochemical test results. Results: PKU patients do not have a higher incidence of overweight and obesity than healthy controls, the proportion of energy derived from carbohydrates in their diets was below the recommended level, and their total energy intake was below the recommended daily allowance. It was remarkable that patients with a treatment adherence ratio of <50% displayed a higher prevalence of overweight and abdominal obesity in comparison to those with a more favorable treatment adherence ratio. Conclusions: In view of the growing prevalence of overweight in the general population, PKU patients should be kept under close long-term follow-up. Particularly in the group with low treatment compliance, more caution should be taken in terms of adverse outcomes.

## 1. Introduction

Phenylketonuria (PKU; OMIM 261600) is a hereditary metabolic disorder characterized by a deficiency of the enzyme phenylalanine hydroxylase, which converts the amino acid phenylalanine to tyrosine [1]. The primary treatment for PKU is a dietary regimen restricted in phenylalanine (Phe), with the supplementation of essential amino acids and micronutrients [2]. The exclusion of protein-rich natural foods and inclusion of low-Phe substitutes may give rise to an imbalanced diet, increasing the risk of overweight and obesity; the prevalence of and propensity for excess adiposity in PKU is a cause of concern [3]. 

Since 1975, the global prevalence of overweight has almost tripled [4]. The World Health Organization (WHO) describes overweight and obesity as abnormal or excessive accumulation of fat. Abdominal obesity is related to dyslipidemia, hypertension, insulin resistance, and inflammation. These alterations are, in the long run, associated with non-communicable diseases such as cardiovascular diseases, non-insulin-dependent diabetes mellitus, musculoskeletal disorders, pulmonary diseases, and cancer [5,6]. 

A substantial body of evidence from numerous studies indicates that patients diagnosed with PKU exhibit a higher body mass index (BMI) compared to the general population [7,8,9,10]. Some studies have demonstrated that the BMI and fat mass of female patients diagnosed with PKU are higher than those of control groups [1,11]. A study conducted in the United States revealed that patients with PKU in the pediatric age group were more likely to be obese or overweight [7]. In a Spanish study that evaluated patients with PKU during adolescence, it was found that their BMI was higher than that of healthy controls [12]. A gradual increase in the frequency of overweight and obesity was observed among patients with PKU living in the same region in Brazil at two-year intervals. It has been proposed that non-adherence to nutritional therapies by patients may also be a contributing factor to the elevated risk of overweight [13].

However, the results of all studies in which anthropometric evaluations of PKU patients were performed are not consistent with one another. Mazzola and colleagues observed that 75% of the cohort of patients with PKU exhibited a normal BMI, while 22% displayed an elevated BMI [14]. Other studies have demonstrated no significant difference between patient groups and healthy controls in terms of BMI values [15,16,17,18,19,20,21].

Sailer et al. reported that although BMI and waist circumference z-score values were increased in the patient group compared to the control group, height z-scores were lower than in the control group [22]. While some studies did not report a negative effect on growth, other studies indicated that growth was retarded, including low height z-scores [16,20,23,24].

In patients with phenylketonuria, some studies have reported higher fat mass in all patients, with a particularly pronounced effect observed in female subjects [11]. In other studies, fat mass was found to be significantly increased in male patients [22]. In contrast to these findings, some studies demonstrated no significant difference between the patient and control groups [14,16,18,20].

In their study on the nutritional composition of low-protein food products used in the dietary management of PKU, Pena and colleagues observed that the majority of these products exhibited higher energy and carbohydrate content, and that over half of them had a higher fat content than typical foods [25]. Furthermore, the quality of fat and fiber used in these products differs from that of normal foods [26]. Excessive consumption may result in deficiencies of micronutrients and excessive energy intake due to the deficiency of micronutrient sources [22]. It is proposed that a reduction in protein intake resulting from the implementation of specific nutritional therapies in the context of amino acid metabolism disorders may precipitate the consumption of carbohydrate- and fat-rich foods, thereby increasing the likelihood of weight gain [22]. Furthermore, it is important to consider the differences in the absorption and bioavailability of synthetic amino acid products used in hereditary disorders of amino acid metabolism compared to natural protein sources, as this affects their suitability as protein alternatives [18,27].

The present study aims to evaluate the body composition and nutritional biochemical biomarkers in adult patients with PKU who are on Phe-restricted and essential amino acid-supplemented nutrition therapy and to investigate the relationships, if any, between these parameters and patient gender, adherence to dietary therapy, and disease type, defined as mild or classic PKU.

## 2. Materials and Methods

This single-center, cross-sectional, observational, and descriptive study was conducted in the Division of Pediatric Nutrition and Metabolism, Istanbul Faculty of Medicine, Istanbul University, between October 2023 and June 2024. The study was approved by the Istanbul Faculty of Medicine Ethics Committee (file number: 2023/1636). Patients with mild or classic PKU, aged 18 years or older, admitted to the outpatient clinic during the study period, and who were willing to participate in the study were selected. Written informed consent was obtained from all volunteers or their legal guardians after the study procedure was explained.

### 2.1. Patient and Control Groups

Patients with Phe levels of >1200 µmol/L at the time of initial diagnosis or a Phe tolerance of <20 mg/dL on follow-up were classified as classic PKU, and those with Phe levels of 600–1200 µmol/L at the time of initial diagnosis or a Phe tolerance of >20 mg/dL on follow-up were classified as mild PKU. Time of diagnosis was defined as early if patients were diagnosed before the age of 30 days and late if diagnosed at 30 days or older. All patients were on nutrition treatment after diagnosis, which consisted of a low-Phe diet and a Phe-free L-amino acid mixture. As a constituent of nutritional treatment, patients were consuming food substitutes that were free of phenylalanine. No patient had received tetrahydrobiopterin treatment before or during the study. Patients with other conditions that might interfere with normal physical development were excluded from the study. The control group consisted of healthy, age- and sex-matched siblings of the patients with inborn errors of metabolism.

### 2.2. Data Collection

The data collected from all volunteers included diagnosis, age, sex, anthropometric measurements, bioelectrical impedance analysis results, current food intake records, and biochemical test analyses. To assess treatment adherence, Phe levels determined from dried blood spot samples over the previous two years were recorded.

### 2.3. Anthropometric Measurements 

Anthropometric measurements including body weight and standing height, body analysis measurements, and biochemical tests were conducted in the morning. The anthropometric measurements were performed without shoes and while wearing light clothing according to standard techniques [28]. BMI was calculated as weight (kg)/height^2^ (m). A BMI of <18.5 indicated underweight, 18.5–24.9 indicated normal, 25–29.9 indicated overweight, and >30 indicated obesity [29]. Waist circumference was measured in a standing position, midway between the lower rib margin and the iliac crest, at the end of a normal exhalation, to the nearest 1 mm. The upper limit of the normal range for waist circumference was considered the 90th percentile [30]. The waist circumference values of the adults were evaluated using the reference values for Turkish adults reported by Sonmez et al. [31].

### 2.4. Bioelectrical Impedance Analysis 

Body composition was evaluated indirectly by bioelectrical impedance analysis (BIA) using an InBody 230, Biospace (Biospace Co., Ltd., Seoul, South Korea) device. The participants were asked to avoid exercise before the analysis. Measurements were taken while the participants were barefoot and with all metal garnitures removed. Lean body mass (skeletal muscle mass, SMM) and fat mass (FM) were measured in kilograms, and the waist-to-hip ratio (WHR) was calculated.

### 2.5. Dietary Assessment

The dietary data of the patients were obtained from three-day food intake records and analyzed using BeBis dietary analysis software (8th version, Umraniye, Turkey). The daily consumption of natural protein, special medical food-derived protein, fat, carbohydrate, and energy intakes were determined. The medical food formulas consumed by the patients were defined as mixtures of synthetic essential amino acids. 

### 2.6. Biochemical Analysis 

Blood glucose, uric acid, AST, ALT, total cholesterol, HDL cholesterol, LDL cholesterol, triglyceride, albumin, hsCRP, HbA1c, and serum insulin concentrations were measured automatically using conventional methods. HOMA-IR was calculated by multiplying fasting glucose (mmol/L) by fasting plasma insulin (μU/mL), then dividing by the constant 22.5. The cut-off point for HOMA-IR indicating insulin resistance was considered to be 2.5 [32].

### 2.7. Treatment Adherence 

The treatment target for patients with PKU was set at a Phe level of 60–600 µmol/L. The treatment adherence ratio was defined as the number of age-appropriate target blood Phe level measurements in one calendar year divided by the number of all blood Phe measurements performed that year.

### 2.8. Statistical Analysis 

Descriptive statistics included mean, standard deviation, median, minimum and maximum, frequency, and ratio. We used the Shapiro–Wilk test, coefficient of variation, histogram plots, detrended normal plots, skewness, and kurtosis measures to check for normal distribution. In cases where both of the variables were quantitative, we applied the Student *t*-test if the quantitative variables were normally distributed, and the Mann–Whitney U test if at least one of the variables was not normally distributed. The Pearson Chi-squared test was used when both variables were qualitative and Fisher’s exact test was used if the assumptions of the Chi-square test were not met. To assess correlations between quantitative variables, we used Pearson’s correlation for normally distributed data and Spearman’s rho otherwise.

## 3. Results

### 3.1. Patient Characteristics

The study group comprised 37 patients with PKU and the control group included 26 age- and sex-matched healthy siblings. In the patient and control groups, 13 (35%) and 15 (58%) were females, respectively. The mean age of the study group was 24.6 ± 5.4 (median: 23.3; range: 18.5–41.3) years and the mean age of the control group was 24.2 ± 4.4 (median: 23.9; range: 18–32.3) years. There were no statistically significant differences in age (*p* = 0.074) or gender (*p* = 0.076) between the groups (Appendix A). The study cohort included 27 (73%) classic PKU and 10 (27%) mild PKU patients. Twenty-four patients (65%) were diagnosed through newborn screening, whereas the remaining patients were late-diagnosed cases. The mean treatment adherence ratio of the patients in the last 2 years before inclusion in the present study was 39% ± 39% (median: 28%; range: 0–100%). A total of 13 patients exhibited a treatment adherence ratio of 50% or greater. 

The distribution of demographic, anthropometric, and biochemical analysis data in the study and control groups are presented in Appendix A. The demographic, anthropometric, and biochemical data of the study group were found to be comparable to those of the control group, with the exception of the albumin levels. A statistically significant difference was observed in albumin levels (*p* = 0.001) between the PKU group and the control group, with the former exhibiting higher levels (Appendix A). 

### 3.2. Comparison of Male and Female PKU Patient Data

A comparison of the data for males and females in the PKU group revealed that, although the mean height and weight of males with PKU were higher, only the height exhibited a statistically significant difference (*p* = 0.000). The mean percentage of SMM in male PKU patients was found to be statistically higher (*p* = 0.000), while the mean percentage of fat mass was observed to be lower (*p* = 0.000). The mean values of uric acid (*p* = 0.020) and albumin (*p* = 0.018) were found to be statistically significantly lower in females with PKU than in males. The rest of the biochemical analysis results did not differ between genders (Table 1). A review of the three-day food intake records of the study group disclosed that the mean daily calorie consumption of the male patients with PKU was statistically significantly higher than that of the female patients with PKU (*p* = 0.000). The proportion of calories from protein (*p* = 0.042) in the diet of females with PKU was statistically significantly higher than that of males and the proportion of calories from carbohydrates was statistically significantly lower (*p* = 0.003). Despite a greater proportion of calories from fat in the dietary intake of female patients compared to their male counterparts, no statistically significant difference was identified (Table 2).

### 3.3. Comparison of the Data Collected from PKU Patients and Control Group, Stratified by Gender

The mean weight of male patients with PKU was statistically significantly lower than that of healthy male controls (*p* = 0.047). Although not statistically significantly different, mean height, WHR, and fat mass percentage were lower and SMM percentage was higher in males with PKU compared to healthy men. BMI and waist circumference were not statistically significantly different between the two groups. Mean uric acid and albumin levels were statistically significantly lower in males with PKU compared to controls (*p* = 0.005/*p* = 0.003). Further statistical analysis revealed no other significant differences in the biochemical analyses (Table 1).

There were no statistically significant differences in height, weight, WHR, SMM percentage, and fat mass percentage between females with PKU and healthy female controls. The mean values of ALT and AST were statistically significantly higher in females with PKU compared to controls, although ALT and AST values were within the reference range. No statistically significant differences were observed in the remaining biochemical parameters (Table 1).

t: *t*-test/m: Mann–Whitney U test/X^2^: Chi-square test. ALT: alanine aminotransferase; AST: aspartate aminotransferase; CRP: C-reactive protein; BMI: body mass index; HbA1c: hemoglobin A1c; HOMA-IR: homeostasis model assessment—insulin resistance; PKU: phenylketonuria; SD: standard deviation; SMM: lean muscle mass; WHR: waist-to-hip ratio. *p* < 0.05 is considered to be statistically significant. ^1^ *p*-values determined by comparing data from male patients and control group, ^2^ *p*-values determined by comparing data from female patients and control group, ^3^ *p*-values determined by comparing data from male and female patient groups.

### 3.4. Comparison of Data between Patients Diagnosed with Mild PKU and Classical PKU

Compliance with treatment of patients with classic PKU was statistically significantly lower than that of patients with mild PKU (*p* = 0.023). The analysis revealed no statistically significant difference in anthropometric measurements between patients with mild PKU and those with classic PKU. With the exception of CRP, the biochemical analysis parameters were not statistically different. The mean CRP level was observed to be statistically significantly higher in the classic PKU group relative to the mild PKU group; however, the mean CRP level remained within the reference range in both groups (Table 3). There were no statistically significant differences in three-day food intake record analyses between the classic and mild PKU patients in the study group, but the natural protein consumption, natural protein/synthetic amino acid mixture ratio, and amount of protein consumed per 100 kcal were lower, while synthetic essential amino acid and energy consumption were higher in classic PKU patients (Table 4). 

### 3.5. Comparison of Data between Patients with Treatment Adherence Ratio ≥ 50% and <50%

Among patients diagnosed with PKU, mean weight, BMI, WHR, and fat mass percentage were statistically significantly lower, while SMM percentage was statistically significantly higher in patients with a treatment adherence ratio of ≥50% compared to patients with a treatment adherence ratio of <50% (Table 5). When correction was made for the effect of disease classification, mean weight (*p* = 0.007), BMI (*p* = 0.004), WHR (*p* = 0.011), fat mass (*p* = 0.005), and fat mass percentage (*p* = 0.014) were statistically significantly lower, while SMM percentage (*p* = 0.039) was statistically significantly higher. The analysis of the food consumption records over a three-day period revealed no statistically significant differences between the group with a treatment adherence ratio < 50% and the group with a treatment adherence ratio ≥50%. However, it was observed that the quantity of natural protein intake per kilogram, total protein consumed per kilogram, and protein intake per calorie were higher in the cohort whose treatment compliance was greater (Table 6).

### 3.6. Correlation Analysis of Patient Data with Treatment Compliance

A negative correlation was observed between the ratio of treatment adherence of the study group and weight (Spearman’s rho = −0.550), BMI (Spearman’s rho = −0.411), and waist circumference (Spearman’s rho = −0.485) measurements.

## 4. Discussion

Nutritional therapies remain the primary treatment option for many inherited metabolic disorders including PKU [33]. It has long been postulated that the Phe-restricted diet consumed by patients with PKU, which necessitates the extensive use of low-protein energy-rich food products, may result in an increased consumption of energy and an imbalance in the diet [34]. This, in turn, may give rise to an increased susceptibility to abdominal obesity and metabolic syndrome in this patient population over the long-term course of dietary treatment. Consequently, studies conducted with different cohorts in various countries have yielded findings that are not fully aligned with one another on this issue [21,35,36,37,38]. The principal aim of this study was to assess the impact of nutritional therapy on body composition and biochemical markers in adult patients with PKU. Furthermore, we investigated the potential associations between these parameters and demographic and clinical factors, including patient gender, adherence to dietary therapy, and PKU type as mild and classic.

The main findings of this study indicate that patients with PKU do not have a higher incidence of overweight and obesity than healthy controls, the proportion of energy derived from carbohydrates in their diets was below the recommended level, and their total energy intake was below the recommended daily allowance (RDA). Our findings further illustrated that male patients with PKU exhibited statistically significantly lower body weight than control subjects, whereas anthropometric evaluations of female patients with PKU demonstrated body compositions that were comparable to those of healthy female controls. Finally, it was notable that patients with a treatment adherence ratio of <50% displayed a statistically higher prevalence of overweight and abdominal obesity in comparison to those with a more favorable treatment adherence ratio. 

BMI assessments of individuals with PKU and healthy controls were not statistically different. Moreover, among PKU patients, a low BMI was more common than in healthy controls (16.2–3.8%). In addition, while the rate of being overweight and obese among controls was 50% in total, this rate was 32.5% in PKU patients. Although other studies have found higher rates of overweight and obesity in PKU patients compared with healthy controls, it was noteworthy that they were lower in our study group [7,8,11]. Some studies have reported no changes in BMI and prevalence of overweight and obesity between patients with PKU and healthy subjects [15,18,20]. Similarly, a previous meta-analysis suggested no significant association between PKU and obesity [34]. 

There was no significant change in the biochemical parameters of PKU patients in accordance with metabolic syndrome and insulin resistance compared to healthy participants. Two studies reported significantly increased total cholesterol and LDL/HDL ratio and lower HDL levels in patients with PKU versus controls [11,39]. On the contrary, although there was no statistical difference, total cholesterol and LDL cholesterol levels were lower and HDL cholesterol levels were higher in our patient group compared to the control group. Another study also revealed lower plasma levels of total cholesterol and LDL cholesterol compared to healthy controls [37]. The reason for the lack of difference in our patient group was assumed to be due to the low total amount of energy consumed by the patients that was below the RDA and the mean proportion of energy derived from carbohydrate being 55.9%. We observed that our patient group did not consume a carbohydrate-rich diet compared to the other study groups, although a Phe-restricted diet may be characterized by greater carbohydrate consumption than the general population [40,41].

The mean albumin level was found to be statistically significantly higher in patients with PKU than in healthy controls. It can reasonably anticipated that, in a group receiving phenylalanine-restricted special nutritional therapy, the ratio of energy provided from carbohydrates will be higher and the ratio of energy provided from protein will be lower compared to the RDA due to this nutritional therapy. In the long term, this type of diet therapy, which is enriched with carbohydrates and low in protein content, may have adverse effects on growth, development, and the synthesis of body proteins, decrease muscle mass, and increase fat mass and the incidence of metabolic syndrome in the patient group. However, in both the male and female patient groups, the ratio of energy derived from carbohydrates was below the levels recommended by the RDA, with the average ratio of energy derived from protein being between 10 and 12.5%. It was hypothesized that this was the reason why the mean albumin level was statistically higher in the patient group.

In a study by Barta et al. using multifrequency BIA, female (but not male) adult subjects with PKU had increased FM and decreased muscle mass as compared to healthy controls [36]. In contrast, Alghamdi et al. observed no statistically significant difference in body composition in adult patients [41]. In our cohort, though not statistically significant, muscle mass was higher and fat mass was lower than controls. Despite the absence of a statistically significant difference, the WHR values observed in the patient group were also found to be lower than those observed in the control group. This was attributed to the fact that the energy intake was lower than the RDA and the carbohydrate intake was lower than recommended. Similar to our results, the other groups reported no differences in FM evaluated by BIA in patients with PKU compared with matched or age-related controls [20,35]. Some of the studies that assessed muscle mass found that muscle mass was normal in people diagnosed with PKU [42], while others reported a loss of muscle mass [43,44].

Female patients with PKU have a higher BMI and prevalence of overweight compared with the general population in a number of retrospective studies [7,45,46]. In our patient group, anthropometric and biochemical evaluations of females with PKU were similar to those of healthy controls, whereas the mean weight of males with PKU was statistically significantly lower than that of healthy men. The proportion of overweight and obese individuals was lower in both male and female PKU groups compared to control groups of the same sex. Individuals with PKU were less prone to obesity than the control group. Among male PKU individuals, those with low BMI were more than controls. In this respect, we suggest that especially male PKU patients should be carefully evaluated in terms of nutritional deficiencies. 

The mean values for height, muscle mass, and fat mass exhibited statistically significant differences between male and female patient groups. In our cohort, males were taller and had more muscle mass and less fat mass than females, which was expected due to gender differences. Within another PKU cohort, adolescent and adult males reported higher SMM than females [46]. No statistically significant difference was observed in waist circumference and BMI between the two genders. Consequently, it was hypothesized that there was no difference in the prevalence of abdominal obesity between male and female patients. 

The results of our study indicate that there was no statistically significant difference between the anthropometric measurements and nutritional biochemical parameters of patients with classical PKU and those with mild PKU. However, in another study, the BMI values of patients with classical PKU were observed to be higher than those of the control group. It has been proposed that this may be due to increased energy consumption in patients with classical PKU with the intention of preventing an increase in Phe levels [34]. In our patient group, the analyses of three-day food intake records in classical PKU patients revealed no statistically significant difference from those of mild PKU patients. 

In the patient cohort, the primary factor contributing to abdominal obesity was not the disease classification or the gender of the patients, but compliance with treatment. In various studies, a positive correlation of mean Phe levels with BMI [1,39], as well as between mean Phe levels and the prevalence of overweight, was demonstrated, suggesting the association of good metabolic control with a reduced risk of overweight [7,21]. Conversely, two studies from Spain reported a higher prevalence of overweight and BMI in patients with good metabolic control compared with those with poor metabolic control [47,48]. 

Although a lifetime evaluation will provide more detailed information regarding the impact of dietary treatment on anthropometric measurements, such as height, weight, BMI, WHR, and waist circumference, evaluating the two-year treatment adherence may also be informative in reaching similar conclusions. No statistically significant differences were observed between the three-day food consumption records of the two groups with different treatment adherence ratios. Nevertheless, the quantity of natural protein intake per kilogram, total protein consumed per kilogram, and protein intake per calorie were observed to be higher in the cohort exhibiting better treatment compliance, although there was no statistically significant difference. The ratio of total energy consumed to the RDA was similar in both groups. This indicates that an elevated protein intake per calorie in the cohort with enhanced treatment efficacy may diminish the propensity for obesity. The quality and quantity of natural protein are important factors in determining SMM. Evans et al. showed in a small cohort of children with PKU that natural protein intake above 0.5 g/kg/day was associated with enhanced body composition [18]. Furthermore, Huemer et al. reported that natural protein intake correlated with SMM [19]. In a study by Rocha et al., no significant difference was detected in the median intake of natural protein and protein substitutes between individuals with overweight/obesity and others with PKU [49].

The present study did not evaluate the physical activity levels of the participants. Additionally, the food consumption of the patients was evaluated on a cross-sectional basis. The potential for patients to alter their dietary habits during the assessment periods of food consumption, coupled with the fact that the cross-sectional evaluation does not reflect long-term dietary habits, may have precluded the identification of differences in food intake between patient groups. Furthermore, the analysis of patients’ three-day food intake records based on self-reported data may not accurately reflect their actual dietary intake as patients tend to report dietary intakes that align with the prescribed recommendations rather than their actual intake.

Pugliese et al. have pointed out that there is considerable heterogeneity in the outcomes reported in the PKU literature [50]. The reasons for this are multifaceted. The assessment and management of patients with PKU is challenging, with significant differences in PKU management practices between countries [51]. The heterogeneity of PKU mutations has consequences for both biochemical and metabolic phenotypes, leading to variability in treatment needs. Furthermore, patients with PKU may suffer from functional and neuropsychological disabilities which impact their adherence to dietary treatments. Caution should be exercised in interpreting the results of studies that employ different criteria for classifying overweight and obese individuals. In addition, the assessment of vitamin, mineral, and trace element intakes, together with analysis of energy, carbohydrate, and fat ratios in dietary intake, may be useful in explaining the differences in the prevalence of obesity and metabolic syndrome in patients.

## 5. Conclusions

The present study revealed that long-term, Phe-restricted nutrition therapy did not cause any adverse biochemical changes in body composition in our cohort. In view of the growing prevalence of overweight in the general population, patients with PKU should be kept under close long-term follow-up. Body composition profiling plays a pivotal role in nutritional evaluation and in understanding the efficacy of nutritional therapy. This is crucial for the prevention of overweight and obesity, as well as the associated comorbidities. Particularly in the group with low treatment compliance, more caution should be paid in terms of adverse outcomes. To properly address problems and to assist in guiding the clinical practice of health professionals, future studies must employ improved methodology.

## Figures and Tables

**Table 1 nutrients-16-03355-t001:** Comparison of demographic, anthropometric, and biochemical data between male and female patients in the study group vs. control group.

		Control Group—Males			PKU Group—Males			*p* ^1^	Control Group—Females			PKU Group—Females			*p* ^2^	*p* ^3^
		Mean ± SD		Median	Mean ± SD		Median		Mean ± SD		Median	Mean ± SD		Median		
Age (year)		25.8 ± 4.0		27.7 (20.0–30.8)	24.0 ± 5.6		22.0 (18.5–41.3)	0.186 ^m^	23.0 ± 4.4		21.7 (18.0–32.3)	25.8 ± 5.0		24.0 (19.5–34.7)	0.135 ^t^	0.150 ^m^
Participants (*n*)		11			24				15			13				
Treatment compliance (%)					45.7 ± 40.7		37.5 (0–100)					29.0 ± 35.1		10.0 (0–100)		0.387^m^
BMI *n* (%)	Underweight	0			5	20.8		0.351 ^X^2^^	1	6.7		1	7.7		0.630 ^X^2^^	0.262 ^X^2^^
	Normal	5	45.5		11	45.8			7	46.6		8	61.5			
	Overweight	4	36.4		6	25.0			4	26.7	9	1	7.7			
	Obese	2	18.1		2	8.4			3	20.0		3	23.1			
Waist circumference (%)	Normal	9	81.9		21	87.5		0.640 ^X^2^^	11	73.3		8	61.5		0.665 ^X^2^^	0.190 ^X^2^^
	Increased	2	18.1		3	12.5			3	20.0		4	30.8			
	No data	0			0				1	6.7	786	1	7.7			
Height (cm)		173.2 ± 8.2		172.0 (158.0–187.0)	171.7 ± 6.7		171.5 (161.0–189.0)	0.576 ^t^	160.8 ± 5.2		160.0 (155.0–171.0)	160.2 ± 6.9		159.0 (152.0–173.0)	0.783 ^t^	0.000 ^t^
Weight (cm)		78.8 ± 12.5		75.2 (63.2–95.9)	68.1 ± 13.2		64.6 (47.9–93.8)	0.047 ^t^	64.2 ± 12.5		63.7 (41.8–86.4)	63.6 ± 14.8		57.0 (46.3–89.2)	0.821 ^m^	0.236 ^m^
WHR		0.91 ± 0.03		0.90 (0.89–0.95)	0.86 ± 0.06		0.87 (0.74–0.95)	0.166 ^t^	0.89 ± 0.06		0.89 (0.77–0.95)	0.89 ± 0.07		0.88 (0.78–1.07)	0.905 ^t^	0.274 ^t^
SMM percentage (%)		41.9 ± 3.9		42.5 (35.1–48.2)	42.3 ± 5.0		42.3 (34.8–49.7)	0.816 ^t^	35.3 ± 3.6		35.0 (29.2–41.6)	33.9 ± 4.3		33.0 (26.2–42.7)	0.372 ^t^	0.000 ^t^
Fat percentage (%)		25.5 ± 6.5		23.9 (14.2–34.7)	23.7 ± 8.8		24.9 (10.0–37.6)	0.547 ^t^	34.5 ± 7.1		33.8 (20.6–46.0)	36.5 ± 8.3		36.4 (20.7–51.3)	0.522 ^t^	0.000 ^t^
Glucose (mg/dL)		87.2 ± 6.7		90.0 (75.0–93.0)	85.5 ± 13.9		81.5 (67.0–126.0)	0.113 ^m^	84.1 ± 7.7		84.0 (72.0–99.0)	81.6 ± 8.0		80.5 (69.0–95.0)	0.492 ^t^	0.784^m^
Uric acid (mg/dL)		5.7 ± 1.2		5.6 (3.9–7.7)	4.4 ± 0.6		4.5 (3.3–5.3)	0.005 ^t^	3.5 ± 0.8		3.8 (1.8–4.8)	3.6 ± 0.9		3.9 (2.3–0.49)	0.806 ^t^	0.020 ^t^
AST (U/L)		26.3 ± 11.5		21.7 (13.2–53.7)	20.7 ± 4.5		20.2 (14.0–30.0)	0.146 ^t^	15.2 ± 2.9		14.8 (10.7–19.8)	18.5 ± 4.5		17.8 (13.8–28.0)	0.040 ^t^	0.221 ^t^
ALT (U/L)		32.0 ± 18.1		33.6 (5.5–65.0)	21.4 ± 7.6		18.7 (10.0–41.0)	0.087 ^t^	11.5 ± 4.7		10.7 (5.5–21.6)	18.2 ± 6.8		17.9 (9.9–28.0)	0.009 ^t^	0.261 ^t^
Total cholesterol (mg/dL)		194.1 ± 48.4		179.0 (131.0–307.4)	163.7 ± 34.8		177.0 (113.0–217.0)	0.074 ^t^	164.4 ± 57.2		173.0 (21.6–234.0)	165.1 ± 25.6		157.0 (127.0–220.0)	0.971 ^t^	0.919 ^t^
Triglyceride (mg/dL)		142.2 ± 81.4		116.6 (61.8–277.3)	109.5 ± 41.3		99.4 (57.0–187.8)	0.540 ^m^	83.6 ± 22.9		81.4 (46.5–131.5)	91.5 ± 40.8		79.0 (35.0–160.4)	0.606 ^t^	0.311 ^t^
HDL cholesterol (mg/dL)		45.0 ± 15.1		42.2 (31.0–82.7)	48.5 ± 10.7		49.0 (33.0–68.0)	0.180 ^m^	54.9 ± 16.9		53.7 (21.6–86.2)	52.4 ± 16.5		47.5 (34.0–85.1)	0.730 ^t^	0.488 ^t^
LDL cholesterol (mg/dL)		118.6 ± 33.4		106.0 (80.0–198.0)	103.8 ± 34.6		112.0 (56.0–175.0)	0.287 ^t^	100.4 ± 28.2		108.0 (50.0–142.0)	96.2 ± 19.4		89.0 (79.0–137.0)	0.700 ^t^	0.555 ^t^
Albumin (g/dL)		5.0 ± 0.1		5.0 (4.8–5.1)	5.7 ± 2.3		5.2 (4.7–15.6)	0.003 ^m^	4.8 ± 0.3		4.8 (4.3–5.2)	4.9 ± 0.2		5.0 (4.6–5.3)	0.234 ^t^	0.018 ^m^
CRP (mg/L)		2.0 ± 1.4		1.7 (0–4.0)	1.7 ± 1.7		1.3 (0–5.0)	0.392 ^m^	2.3 ± 3.9		0.7 (0–14.0)	1.4 ± 0.6		1.4 (1.0–2.0)	0.477 ^m^	0.785 ^m^
Insulin (µU/mL)		12.0 ± 8.0		9.5 (3.8–29.5)	19.2 ± 24.3		9.8 (4.1–85.9)	0.606 ^m^	10.5 ± 5.9		8.6 (4.2–23.0)	28.1 ± 25.2		16.0 (1.0–57.0)	0.195 ^t^	0.267 ^m^
HbA1c (mmol/mol)		5.1 ± 0.3		5.1 (4.9–5.6)	5.0 ± 0.4		5.0 (4.2–5.7)	0.330 ^m^	5.1 ± 0.4		5.1 (4.1–5.8)	4.8 ± 0.3		4.7 (4.4–5.2)	0.120 ^t^	0.309 ^t^

t: *t*-test/m: Mann-whitney u test/X^2^: Chi-square test. ALT: Alanine aminotransferase; AST: Aspartate aminotransferase; CRP: C reactive protein; BMI: Body mass index; HbA1c: Hemoglobin A1c; p: *p*-value; PKU: Phenylketonuria; SD: Standart deviation; SMM: Lean muscle mass; WHR: Waist to hip ratio *p* < 0.05 is considered to be statistically significant. ^1^ *p*-values determined by comparing data from male patient and control groups, ^2^ *p*-values determined by comparing data from male patient and control groups, ^3^ *p*-values determined by comparing data from male and female patient groups.

**Table 2 nutrients-16-03355-t002:** Comparison of nutritional data between male and female PKU patients.

	Male Patients		Female Patients		*p*
	Mean ± SD	Median	Mean ± SD	Median	
Natural protein (g)	25.8 ± 6.7	25.0 (14.4–43.8)	22.7 ± 8.1	20.0 (14.0–36.3)	0.219 ^t^
Natural protein (g/kg)	0.4 ± 0.1	0.4 (0.2–0.8)	0.4 ± 0.2	0.3 (0.2–0.7)	0.734 ^t^
EAA (g)	25.5 ± 7.6	26.4 (14.0–40.0)	26.7 ± 6.7	24.0 (17.5–40.0)	0.758 ^t^
EAA (g/kg)	0.4 ± 0.1	0.4 (0.2–0.8)	0.4 ± 0.1	0.4 (0.3–0.6)	0.295 ^t^
Total protein (g)	51.4 ± 10.4	50.8 (33.8–73.8)	48.9 ± 9.9	49.8 (34.0–65.0)	0.494 ^t^
Total protein (g/kg)	0.8 ± 0.2	0.8 (0.4–1.3)	0.8 ± 0.2	0.8 (0.4–1.3)	0.727 ^t^
Natural protein/EAA	1.1 ± 0.4	1.0 (0.5–2.1)	0.9 ± 0.5	0.7 (0.4–2.0)	0.239 ^t^
Protein (%)	10.4 ± 2.7	10.0 (7.0–19.0)	12.6 ± 3.2	13.0 (9.0–18.0)	0.042 ^m^
Carbohydrate (%)	58.4 ± 6.4	58.5 (48.0–70.0)	51.5 ± 5.7	52.0 (42.0–66.0)	0.003 ^t^
Fat (%)	31.3 ± 6.3	31.0 (16.0–41.0)	35.4 ± 5.5	36.0 (25.0–44.0)	0.056 ^t^
Energy (kcal)	2072 ± 317	2078 (1500–2600)	1619 ± 247	1565 (1200–2100)	0.000 ^t^
Energy/RDA energy (%)	82.5 ± 15.5	79.2 (51.7–126.3)	79.6 ± 12.2	79.0 (57.1–105.0)	0.570 ^t^
Total protein/100 kcal	2.5 ± 0.7	2.4 (1.7–4.9)	3.1 ± 0.8	3.2 (2.0–4.3)	0.042 ^m^

t: *t*-test/m: Mann–Whitney U test. EAAs: essential amino acids; g: gram; kcal: kilocalories; kg: kilogram; PKU: phenylketonuria; RDA: recommended daily allowance; SD: standard deviation. *p* < 0.05 is considered to be statistically significant.

**Table 3 nutrients-16-03355-t003:** Comparison of demographic, anthropometric, and biochemical data between patients diagnosed with mild PKU and classical PKU.

		Mild PKU			Classical PKU			*p*
		Mean ± SD		Median	Mean ± SD		Median	
Age (year)		24.8 ± 4.5		23.5 (20.4–32.7)	24.5 ± 5.8		26.8 (18.5–41.3)	0.724 ^m^
Gender n (%)	Male	7	70.0		17	62.9		1.000 ^X^2^^
	Female	3	30.0		10	37.1		
Treatment Compliance (%)		60.0 ± 37.0		61.0 (10.0–100)	32.1 ± 37.7		11.0 (0–100)	0.023 ^m^
BMI n (%)	Low	2	20.0		4	14.8		0.770 ^X^2^^
	Normal	5	50.0		14	51.8		
	Overweight	1	10.0		6	22.2		
	Obese	2	20.0		3	11.2		
Waist circumference (%)	Normal	7	70.0		22	81.5		1.000 ^X^2^^
	Increased	2	20.0		5	18.5		
	No data	1	10.0		0	0		
Height (cm)		166.6 ± 6.4		166.5 (157.0–180.0)	168.0 ± 9.4		171.0 (152.0–189.0)	0.468 ^m^
Weight (cm)		66.7 ± 15.0		61.5 (47.9–92.0)	66.4 ± 13.6		63.7 (46.3–93.8)	0.963 ^t^
WHR		0.88 ± 0.05		0.86 (0.79–0.95)	0.87 ± 0.07		0.87 (0.74–1.07)	0.801 ^m^
SMM percentage (%)		39.1 ± 6.9		37.4 (29.8–48.7)	39.4 ± 6.0		38.5 (26.2–49.7)	0.883 ^t^
Fat percentage (%)		28.7 ± 11.9		32.0 (12.3–45.8)	28.0 ± 10.2		29.2 (10.0–51.3)	0.674 ^m^
Glucose (mg/dL)		83.0 ± 12.0		79.0 (69.0–107.0)	84.8 ± 12.9		81.0 (67.0–126.0)	0.747 ^t^
Uric acid (mg/dL)		4.4 ± 0.7		4.6 (3.3–5.3)	4.0 ± 0.8		4.2 (2.3–5.3)	0.314 ^t^
AST (U/L)		20.4 ± 5.2		20.0 (14.0–30.0)	19.8 ± 4.4		19.5 (13.8–28.0)	0.750 ^t^
ALT (U/L)		22.6 ± 9.4		21.8 (11.6–41.0)	19.6 ± 6.6		18.5 (9.9–34.4)	0.508 ^m^
Total cholesterol (mg/dL)		167.0 ± 34.8		171.5 (113.0–217.0)	162.0 ± 30.1		163.5 (117.0–220.0)	0.766 ^t^
Triglyceride (mg/dL)		108.5 ± 44.4		105.5 (57.0–163.0)	99.9 ± 40.7		90.9 (35.0–187.8)	0.834 ^m^
HDL cholesterol (mg/dL)		52.2 ± 13.1		52.8 (35.0–68.0)	48.9 ± 13.2		47.4 (33.0–85.1)	0.490 ^m^
LDL cholesterol (mg/dL)		111.0 ± 34.3		110.0 (60.0–175.0)	95.9 ± 26.7		93.0 (56.0–140.0)	0.291 ^m^
Albumin (g/dL)		5.2 ± 0.2		5.2 (4.9–5.4)	5.6 ± 2.3		5.1 (4.6–5.6)	0.566 ^m^
CRP (mg/L)		0.6 ± 0.7		0.3 (0–2.0)	1.9 ± 1.5		1.4 (0–5)	0.045 ^m^
Insulin (µU/mL)		10.4 ± 1.3		9.8 (9.5–11.8)	24.6 ± 26.2		13.4 (1.0–85.9)	0.611 ^m^
HbA1c (mmol/mol)		4.9 ± 0.2		4.9 (4.7–5.1)	4.9 ± 0.4		4.9 (4.2–5.7)	0.851 ^t^

t: *t*-test/m: Mann–Whitney U test/X^2^: Chi-square test. ALT: alanine aminotransferase; AST: aspartate aminotransferase; BMI: body mass index; CRP: C-reactive protein; HbA1c: hemoglobin A1c; PKU; phenylketonuria; SD: standard deviation; SMM: skeletal muscle mass; WHR: waist-to-hip ratio. *p* < 0.05 is considered to be statistically significant.

**Table 4 nutrients-16-03355-t004:** Comparison of nutritional data between patients diagnosed with mild PKU and classical PKU.

	Mild PKU		Classical PKU		*p*
	Mean ± SD	Median	Mean ± SD	Median	
Natural protein (g)	28.0 ± 8.8	30.0 (16.4–43.8)	23.4 ± 6.3	23.5 (14.0–37.0)	0.086 ^t^
Natural protein (g/kg)	0.5 ± 0.2	0.4 (0.2–0.8)	0.4 ± 0.1	0.4 (0.2–0.7)	0.228 ^t^
EAA (g)	24.3 ± 6.2	23.2 (17.5–35.0)	26.3 ± 7.6	26.4 (14.0–40.0)	0.459 ^t^
EAA (g/kg)	0.4 ± 0.1	0.4 (0.3–0.5)	0.4 ± 0.1	0.4 (0.2–0.8)	0.434 ^t^
Total protein (g)	52.4 ± 11.7	50.7 (39.6–73.8)	49.9 ± 9.7	49.9 (33.8–69.0)	0.513 ^t^
Total protein (g/kg)	0.8 ± 0.3	0.8 (0.5–1.3)	0.8 ± 0.2	0.8 (0.4–1.3)	0.544 ^t^
Natural protein/EAA	1.2 ± 0.5	1.1 (0.6–2.1)	0.9 ± 0.4	0.9 (0.4–2.1)	0.113 ^m^
Protein (%)	11.8 ± 3.6	10.5 (8.0–19.0)	10.9 ± 2.8	10.0 (7.0–18.0)	0.443 ^t^
Carbohydrate (%)	52.9 ± 7.2	53.0 (42.0–68.0)	57.0 ± 6.7	55.0 (47.0–70.0)	0.107 ^t^
Fat (%)	34.5 ± 6.1	36.0 (24.0–43.0)	32.0 ± 6.3	33.0 (16.0–44.0)	0.302 ^t^
Energy (kcal)	1802 ± 349	1682 (1500–2400)	1954 ± 367	2000 (1200–2600)	0.266 ^t^
Energy/RDA energy (%)	81.0 ± 18.7	78.9 (51.7–126.3)	81.6 ± 12.8	78.9 (57.1–105.3)	0.749 ^m^
Total protein/100 kcal	3.0 ± 0.9	2.7 (1.7–4.9)	2.6 ± 0.7	2.3 (1.7–4.3)	0.229 ^m^

t: *t*-test/m: Mann–Whitney U test. EAAs: essential amino acids; g: gram; kcal: kilocalories; kg: kilogram; PKU: phenylketonuria; RDA: recommended daily allowance; SD: standard deviation. *p* < 0.05 is considered to be statistically significant.

**Table 5 nutrients-16-03355-t005:** Comparison of demographic, anthropometric, and biochemical data between patients with treatment adherence ratio ≥ 50% and <50%.

		Treatment Adherence ≥ 50			Treatment Adherence < 50			*p*
		Mean ± SD		Median	Mean ± SD		Median	
Age (year)		25.6 ± 6.5		23.8 (18.5–41.3)	22.1 ± 4.8		22.8 (18.5–33.5)	0.649 ^t^
Gender n (%)	Male	10	76.9		14	58.3		0.305 ^X^2^^
	Female	3	23.1		10	41.7		
BMI n (%)	Low	5	38.4		1	4.3		0.028 ^X^2^^
	Normal	6	46.2		13	54.1		
	Overweight	2	15.4		5	20.8		
	Obese	0			5	20.8		
Waist circumference (%)	Normal	12	92.3		17	73.9		0.382 ^X^2^^
	Increased	1	7.7		6	26.1		
	No data	0			0			
Height (cm)		168.4 ± 7.5		170.0 (154.0–180.0)	167.3 ± 7.5		170.0 (154.0–180.0)	0.720 ^m^
Weight (cm)		58.9 ± 9.9		58.1 (46.3–76.1)	70.6 ± 13.9		66.2 (51.5–93.8)	0.011 ^t^
WHR		0.84 ± 0.05		0.83 (0.76–0.92)	0.89 ± 0.06		0.89 (0.74–1.07)	0.018 ^m^
SMM percentage		42.0 ± 6.1		44.2 (32.5–49.7)	37.9 ± 5.8		37.8 (26.2–49.2)	0.050 ^t^
Fat percentage		22.9 ± 10.1		19.6 (10.0–40.1)	31.1 ± 9.8		30.7 (12.0–51.3)	0.022 ^m^
Glucose (mg/dL)		83.3 ± 9.6		80.0 (77.0–107.0)	84.8 ± 13.9		82.0 (67.0–126.0)	0.735 ^t^
Uric acid (mg/dL)		4.1 ± 0.9		4.3 (2.3–5.3)	4.2 ± 0.8		4.2 (2.4–5.3)	0.904 ^t^
AST (U/L)		19.5 ± 4.7		19.3 (14.0–28.0)	20.2 ± 4.6		19.9 (13.8–30.0)	0.673 ^m^
ALT (U/L)		20.1 ± 8.5		18.4 (10.0–34.4)	20.6 ± 7.0		19.1 (9.9–41.0)	0.870 ^m^
Total cholesterol (mg/dL)		164.2 ± 38.6		162.0 (113.0–217.0)	164.3 ± 29.4		173.0 (117.0–220.0)	0.994 ^m^
Triglyceride (mg/dL)		93.9 ± 45.8		81.2 (35.0–163.0)	105.7 ± 40.5		99.2 (57.0–187.8)	0.555 ^m^
HDL cholesterol (mg/dL)		60.1 ± 11.2		64.5 (43.5–71.0)	46.7 ± 11.9		46.6 (33.0–85.1)	0.022 ^m^
LDL cholesterol (mg/dL)		90.5 ± 28.2		84.5 (60.0–133.0)	104.4 ± 29.9		105.5 (56.0–175.0)	0.328 ^m^
Albumin (g/dL)		5.1 ± 0.2		5.2 (4.6–5.3)	5.7 ± 2.4		5.1 (4.7–15.6)	0.664 ^m^
CRP (mg/L)		0.7 ± 0.9		0.3 (0–2.0)	1.9 ± 1.5		1.4 (0–5.0)	0.102 ^m^
Insulin (µU/mL)		20.6 ± 24.5		10.7 (4.1–57.0)	22.4 ± 0.3		5.0 (4.5–5.5)	0.770 ^m^
HbA1c (mmol/mol)		5.0 ± 0.3		5.0 (4.5–5.5)	4.9 ± 0.4		4.9 (4.2–5.7)	0.693 ^t^

t: *t*-test/m: Mann–Whitney U test/X^2^: Chi-square test. ALT: alanine aminotransferase; AST: aspartate aminotransferase; BMI: body mass index; CRP: C-reactive protein; HbA1c: hemoglobin A1c; PKU: phenylketonuria; SD: standard deviation; SMM: skeletal muscle mass; WHR: waist-to-hip ratio. *p* < 0.05 is considered to be statistically significant.

**Table 6 nutrients-16-03355-t006:** Comparison of nutritional data between patients with treatment adherence ratio ≥ 50% and <50%.

	Treatment Adherence ≥ 50		Treatment Adherence < 50		*p*
	Mean ± SD	Median	Mean ± SD	Median	
Natural protein (g)	25.7 ± 8.5	25.7 (14.0–43.8)	24.1 ± 6.6	23.3 (14.4–37.0)	0.219 ^t^
Natural protein (g/kg)	0.4 ± 0.1	0.4 (0.2–0.8)	0.4 ± 0.1	0.3 (0.2–0.7)	0.734 ^t^
EAA (g)	23.6 ± 7.5	21.0 (14.0–40.0)	26.9 ± 6.9	27.2 (14.0–40.0)	0.758 ^t^
EAA (g/kg)	0.4 ± 0.1	0.4 (0.2–0.8)	0.4 ± 0.1	0.4 (0.2–0.6)	0.295 ^t^
Total protein (g)	49.3 ± 11.5	47.5 (34.0–73.8)	51.2 ± 9.6	51.5 (33.8–69.0)	0.494 ^t^
Total protein (g/kg)	0.8 ± 0.2	0.8 (0.6–1.3)	0.8 ± 0.2	0.8 (0.4–1.3)	0.727 ^t^
Natural protein/EAA	1.2 ± 0.5	1.0 (0.5–2.1)	0.9 ± 0.4	0.9 (0.4–2.0)	0.239 ^t^
Protein (%)	11.2 ± 3.5	10.0 (7.0–19.0)	11.2 ± 2.8	10.5 (7.0–18.0)	0.042 ^m^
Carbohydrate (%)	56.8 ± 7.3	55.0 (47.0–70.0)	55.5 ± 6.9	53.5 (42.0–70.0)	0.003 ^t^
Fat (%)	31.4 ± 6.7	31.0 (16.0–44.0)	33.4 ± 6.0	35.0 (23.0–43.0)	0.056 ^t^
Energy (kcal)	1850 ± 328	1800 (1500–2500)	1946 ± 384	2000 (1200–2600)	0.000 ^t^
Energy/RDA energy (%)	81.1 ± 19.4	78.9 (51.7–126.3)	81.7 ± 11.2	78.9 (57.1–105.2)	0.570 ^t^
Total protein/100 kcal	2.8 ± 0.9	2.7 (1.7–4.9)	2.7 ± 0.7	2.5 (1.7–4.3)	0.042 ^m^

t: *t*-test/m: Mann–Whitney U test. EAAs: essential amino acids; g: gram; kcal: kilocalories; kg: kilogram; RDA: recommended daily allowance; SD: standard deviation. *p* < 0.05 is considered to be statistically significant.

## Data Availability

The data presented in this study are openly available in Zenodo at https:doi.org/10.5281/zenodo.13762514.

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
