# Peer review of "Evaluation of Body Composition and Biochemical Parameters in Adult Phenylketonuria"

_nutrients, 2024, doi:10.3390/nu16193355_

Round 1

Reviewer 1 Report

Comments and Suggestions for Authors

Authors in their article entitled: Evaluation of Body Composition and Biochemical Parameters in Adult PKU have made an effort to answer the question if PKU's high carbohydrate diet causes obesity and therefore the subsequent diseases.  

Their studies are very important from the point of Turkey where the PKU is frequent. Therefore the medical system is very interesting in all studies in this field to reduce the different therapy costs. The author decided to compare the diet and therefore the biochemical parameters of patients with PKU versus healthy people (without PKU). They correctly mention that carbohydrates and fats reach high level feading profiles and are observed for patients with phenylalanine un-toleration. They propose the correct package of blood parameters as well as anthropometric ones. The LHD, HDL total holestereol as wel as triglicerides has been taken into disscusion. The results of their studies disclose no differences between healthy patients and in some cases even better. 

The main explanation of the above is that PKU patients are permanently malnourished. Therefore the over caloletry has not been reached if patients keep in diet restriction. On the other side, the author mentions less musculus in the human body which is derived directly from the lack of proteins in the nutrition. The protein intake decrease is the key for the PKU patients.

From the editorial point the article is well written and readable, moreover, the references are correctly cited.

I recommend putting in the title that these studies are preliminary, due to the small investidated groups. I expect that in Turkey authors can find more than hundreds of people with PKU from different social groups.

Reviewer 2 Report

Comments and Suggestions for Authors

The manuscript nutrients-3234392 presents an original investigation on body composition and nutritional biochemical biomarkers in adult patients with phenylketonuria (PKU). The main aim is the understanding of a potential relationship between Phe-restricted and essential amino acid supplemented nutrition with patient gender, adherence to dietary therapy, and disease severity (mild or classic).

General comments

The paper is of potential interest given the mixed evidence available so far on the topic. It is generally well written, but nonetheless difficult to navigate through due to the lack of sub-paragraphs. Sometimes it is also difficult to distinguish between statistically significant results and observed trends (i.e., not reaching statistically significance thresholds). Therefore, my suggestion is that the authors split the text into multiple sections where they can address specific aims of their research. In my opinion, the main limitation of this work is that the current diet therapy might have a clear relationship with the measured clinical biochemical parameters, while less clear the impact on the body composition, which is the result of many players, time included. If I am correct, the only retrospective analysis was on patient compliance and no other parameters were recorded. How can a single time point analysis be related to a long-lasting process such as the modification of the body composition?

Specific comments

1) The Introduction lacks a description of the current state of the research field with indication of key publications.

2)    Materials and Methods. Please split into sub-paragraphs to improve readability. Please also indicate if patients were consuming phenylalanine-free food substitutes, whose composition might be unbalanced according to current reports.

3)    Results. Please split into sub-paragraphs to improve readability. Make sure to clearly distinguish between statistically significant differences and trends, as specified above.

4)    Lines 128-130. Can the authors elaborate more on that?

5)    Line 155. I believe that 2 p-values are referred to women?

6)    Table 6. There is a discrepancy between the legend and the first row, which reads male and female patients.

7)    Lines 225-226. Please provide suitable references.

8)    Line 310. Please check the appropriateness of the references on “BMI(1, 24)”

9)    Discussion. Again, it is difficult to read due to a mix of statistically significant differences and trends. Please elaborate more on the mechanisms by which adherence to therapy might affect body composition in the long run.

10) Limits of self-reported food intake should be acknowledged.

Comments on the Quality of English Language

The quality of English is generally fine
